# Drug Resistance in Osteosarcoma: Emerging Biomarkers, Therapeutic Targets and Treatment Strategies

**DOI:** 10.3390/cancers13122878

**Published:** 2021-06-09

**Authors:** Claudia Maria Hattinger, Maria Pia Patrizio, Leonardo Fantoni, Chiara Casotti, Chiara Riganti, Massimo Serra

**Affiliations:** 1Pharmacogenomics and Pharmacogenetics Research Unit of the Laboratory of Experimental Oncology, IRCCS Istituto Ortopedico Rizzoli, 40136 Bologna, Italy; claudia.hattinger@ior.it (C.M.H.); mariapia.patrizio@gmail.com (M.P.P.); leonardo.fantoni@ior.it (L.F.); chiara.casotti@ior.it (C.C.); 2Department of Oncology, University of Torino, 10026 Torino, Italy; chiara.riganti@unito.it

**Keywords:** osteosarcoma, drug resistance, personalized medicine, biomarker

## Abstract

**Simple Summary:**

Despite the adoption of aggressive, multimodal treatment schedules, the cure rate of high-grade osteosarcoma (HGOS) has not significantly improved in the last 30 years. The most relevant problem preventing improvement in HGOS prognosis is drug resistance. Therefore, validated novel biomarkers that help to identify those patients who could benefit from innovative treatment options and the development of drugs enabling personalized therapeutic protocols are necessary. The aim of this review was to give an overview on the most relevant emerging drug resistance-related biomarkers, therapeutic targets and new agents or novel candidate treatment strategies, which have been highlighted and suggested for HGOS to improve the success rate of clinical trials.

**Abstract:**

High-grade osteosarcoma (HGOS), the most common primary malignant tumor of bone, is a highly aggressive neoplasm with a cure rate of approximately 40–50% in unselected patient populations. The major clinical problems opposing the cure of HGOS are the presence of inherent or acquired drug resistance and the development of metastasis. Since the drugs used in first-line chemotherapy protocols for HGOS and clinical outcome have not significantly evolved in the past three decades, there is an urgent need for new therapeutic biomarkers and targeted treatment strategies, which may increase the currently available spectrum of cure modalities. Unresponsive or chemoresistant (refractory) HGOS patients usually encounter a dismal prognosis, mostly because therapeutic options and drugs effective for rescue treatments are scarce. Tailored treatments for different subgroups of HGOS patients stratified according to drug resistance-related biomarkers thus appear as an option that may improve this situation. This review explores drug resistance-related biomarkers, therapeutic targets and new candidate treatment strategies, which have emerged in HGOS. In addition to consolidated biomarkers, specific attention has been paid to the role of non-coding RNAs, tumor-derived extracellular vesicles, and cancer stem cells as contributors to drug resistance in HGOS, in order to highlight new candidate markers and therapeutic targets. The possible use of new non-conventional drugs to overcome the main mechanisms of drug resistance in HGOS are finally discussed.

## 1. Introduction

High-grade osteosarcoma (HGOS) is the most common primary malignant tumor of bone for which the identification of new treatment strategies is highly warranted to improve the presently achievable clinical cure rates [1,2].

The most common treatment for primary, conventional HGOS (localized in the extremities, non-metastatic at clinical onset, age lower than 40 years) consists of pre- and post-operative chemotherapy regimens based on different combinations of doxorubicin, methotrexate, and cisplatin with the possible addition of ifosfamide, etoposide and, more recently, liposomal muramyl tripeptide phosphatidylethanolamine (L-MTP-PE; MEPACT) [1,2,3,4,5]. Unfortunately, intensified treatments with these drugs or with the addition of adjuvant agents (i.e., interferons or zoledronic acid), which have been performed in the past decades, have failed to significantly improve cure rates of conventional HGOS patients [1,6,7,8].

Other drugs, such as vincristine, 5-fluorouracil, bleomycin, cyclophosphamide, actinomycin-D, docetaxel, and gemcitabine, among others, proved to be less active and are presently considered only for second-line and rescue chemotherapy protocols, which are far from being standardized [1,9].

The major clinical problem that severely limits the success of HGOS treatment is inherent or acquired drug resistance which, together with development of metastasis, causes the dismal prognosis of relapsed patients, for whom cure probability does not exceed 20–25% [10,11,12]. Therefore, the identification, characterization and clinical validation of drug resistance-related biomarkers is mandatory to indicate new candidate therapeutic targets and novel agents, which may be considered to improve the successful rate of clinical trials for HGOS.

This review is focused on the drug resistance-related biomarkers proven to be of relevant clinical impact or which have recently emerged in HGOS as new candidate therapeutic targets to indicate novel treatment strategies aimed to overcome drug resistance and improve patients’ outcome.

The involvement of the microenvironment, signal transduction pathways or cell cycle and apoptosis disruptions on drug resistance was not considered since it has recently been reviewed in detail [13,14]

## 2. Consolidated and Emerging Drug Resistance-Related Biomarkers

### 2.1. ABC Transporters

Drug resistance remains the most significant obstacle to successful treatment for HGOS, in which one of the main reasons for clinical drug unresponsiveness is the overexpression of the membrane drug transporter ATP binding cassette (ABC) subfamily B member 1 (ABCB1), also known as P-glycoprotein. Several studies have investigated the clinical relevance of ABCB1 expression in HGOS progression, treatment response, or outcome, and the provided results have sometimes been controversial [15,16,17,18,19]. However, different meta-analyses demonstrated that increased ABCB1 protein expression at diagnosis can predict poor survival in HGOS patients [20,21]. Based on this evidence, the overexpression of ABCB1 at diagnosis has been used to design a treatment protocol conducted by the Italian Sarcoma Group, of which the results are presently under revision [5].

Although ABCB1 is a robust negative predictive biomarker in HGOS patients, the analysis of co-expression patterns may lead to the discovery of further biomarkers associated with poor chemosensitivity. For instance, HGOS patients with high expression of both ABCB1 and excision repair cross-complementation group 1 (ERCC1) were shown to have a worse response to neo-adjuvant chemotherapy compared to patients with high ABCB1 overexpression only [22]. This finding is consistent with reduced damage elicited by chemotherapy, due to the simultaneous increased DNA repair and enhanced drug efflux.

Since the role of ABC transporters in HGOS has recently been reviewed [14] it has not been further overviewed in detail in this manuscript.

### 2.2. DNA Repair Factors

Three of the most widely used first-line HGOS chemotherapeutic drugs generate either direct (cisplatin and ifosfamide) or indirect (doxorubicin) DNA damage. This damage is mended by different repair pathways, which include direct repair (DR), nucleotide excision repair (NER), base excision repair (BER), mismatch repair (MMR), single strand break repair (SSBR), double strand break repair (DSBR), nonhomologous end joining (NHEJ) and homologous recombination repair (HRR) [23].

The DNA repair pathways that have most often been linked to drug resistance in HGOS are NER and BER, which may interfere with sensitivity to both first-line and second-line drugs used in this tumor [1,9,24]. Altered expression of different excision repair cross-complementation (ERCC) genes, belonging to the NER pathway, have been reported to be associated with worse prognosis [22] or histological response [25]. Moreover, drugs targeting NER/BER pathways have recently been indicated to be of possible clinical interest to overcome cisplatin unresponsiveness in HGOS [26].

Genetic polymorphisms affecting ERCC1 and excision repair cross-complementation group 2 (ERCC2) genes have been studied and were associated with treatment response and/or survival of HGOS patients [1,24,27].

The expression of apurinic/apyrimidinic exonuclease 1 (APEX1 or APE1), a gene belonging to the BER pathway, was reported to be amplified and overexpressed in HGOS clinical samples [28]. APEX1 increased expression was described to be associated with a trend toward a reduced chemotherapy response and to significantly correlate with development of tumor recurrence and metastasis [28]. Thus, if APEX1 expression level is further confirmed as a prognostic marker for HGOS, it might be considered as a new potential therapeutic target for this tumor.

Another BER gene that has been explored in HGOS is poly(ADP-ribose) polymerase 1 (PARP1). In a recent study performed in a series of relapsed and unresectable HGOS patients, high PARP1 expression showed a clear trend towards an association with worse outcome, but did not reach statistical significance [29]. On the basis of these and other clinical evidence, which showed encouraging activity of the PARP inhibitor olaparib in bone and soft-tissue sarcomas [30], targeting PARP appears to be a promising therapeutic strategy to be further explored. Phase 2 clinical trials with the PARP inhibitor olaparib used alone (ClinicalTrials.gov Identifier: NCT03233204) or in combination with the ataxia telangiectasia and rad3 related (ATR) kinase inhibitor ceralasertib in patients with unresponsive, refractory or recurrent HGOS (ClinicalTrials.gov Identifier: NCT04417062) are presently ongoing. The rationale of the latter combination derives from the fact that ATR is another essential regulator of the DNA damage response, which increased expression and activation was shown to correlate with shorter HGOS patient survival and lower extent of tumor necrosis following neoadjuvant chemotherapy [31]. 

Recently, it has also been demonstrated that nitric oxide (NO) can sensitize cells to DNA damaging drugs through the induction, nitrosation and denaturation of several proteins which are involved in DNA repair, indicating new possible strategies for future therapeutic intervention, despite limitations must still be resolved for clinical applications [32]. Moreover, NO induces the depletion of GSH that generally inactivates platinum(Pt)-based drugs, thus contributing to reverse resistance against DNA damaging agents.

### 2.3. Methotrexate Resistance-Related Factors

Another key agent for HGOS treatment is methotrexate, an antifolate drug, which binds to the dihydrofolate reductase (DHFR) enzyme, leading to inhibition of DNA synthesis and replication, and ultimately causing apoptosis. Increased levels of DHFR have been reported to be one of the major mechanisms responsible for methotrexate resistance in HGOS, along with impaired drug transport due to decreased expression of the membrane-located solute carrier family 19 (folate transporter) member 1 (SLC19A1; also known as reduced folate carrier, RFC or reduced folate carrier 1, RFC1) [33]. In order to overcome these and other common mechanisms of methotrexate resistance, different antifolates have been studied in several experimental models and a few of them have also been evaluated at clinical level, unfortunately without showing activity superior to that exhibited by methotrexate [1,33].

Polymorphisms affecting genes involved in methotrexate transport and metabolism have also been studied and indicated to mediate drug activity and collateral toxicity [27,33], but further investigations and functional analyses are needed before this body of information can effectively be transferred into clinical practice.

### 2.4. Extracellular Vesicles

Several studies performed in different cancers have shown the importance of extracellular vesicles (EVs) in intercellular communication and the interchange of bioactive molecules between tumor and resident cells [34]. These vesicles include exosomes, microvesicles, oncosomes, and microparticles. Recent research findings have strongly supported EVs as key players in mediating drug unresponsiveness and in conferring resistance to drug-sensitive cells in different tumors [34,35,36,37], with increasing evidence also emerging in HGOS.

Different studies have shown that EVs can mediate drug resistance in HGOS through the transfer of coding and non-coding RNAs.

The spread of doxorubicin resistance between different HGOS cell populations was shown to occur through transfer of EVs carrying drug efflux pumps [38]. This study showed that EVs were able to decrease sensitivity to doxorubicin by transferring functional *MDR-1* mRNA (encoding for ABCB1) in MG-63, HGOS cells [38]. The demonstration of EVs release containing *MDR-1* mRNA from HGOS doxorubicin-resistant cells suggested that this may be a mechanism through which resistant tumor cells may spread drug unresponsiveness to sensitive cells, contributing to tumor chemoresistance [38].

Recently, it has been reported that exosomes can mediate chemoresistance of HGOS cells by transmitting circular RNAs (circRNAs) [39]. In this study, the expression of hsa_circ_103801 was found to be upregulated in cisplatin-resistant MG63 cells compared with their drug-sensitive parental cells. Moreover, in the same study, it was found that hsa_circ_103801 was highly present in exosomes derived from cisplatin-resistant cells, and that its spread through these microvescicles was able to decrease the cisplatin sensitivity of MG63 and U2OS HGOS cells, as well as to inhibit apoptosis and increase expression of ABCC1 and ABCB1 [39]. These findings may also indicate exosomal hsa_circ_103801 as a new candidate target to be considered for overcoming HGOS chemoresistance.

In another study [40], dysregulation of miR-25-3p, which inhibits the expression of the *dickkopf WNT signaling pathway inhibitor 3 (DKK3)* gene, was detected in human HGOS tissues and proved to be negatively correlated with clinical outcome. The same authors demonstrated that, in HGOS experimental models, miR-25-3p upregulation promoted tumor growth, invasion, and drug resistance, and that these same effects were also detected after *DKK3* silencing. Interestingly, miR-25-3p was found to be present in tumor-derived exosomes, suggesting that it may exert its oncogenic functions through EVs-mediated dissemination.

Exosome protein cargos were also recently studied by using canine osteosarcoma experimental models and clinical samples [41]. In this study, the protein content of exosomes derived from two canine osteosarcoma, carboplatin-resistant variants (HMPOS-2.5R and HMPOS-10R) was compared to that of their drug-sensitive parental cell line HMPOS. Authors demonstrated that exosomes exhibited distinct protein signatures related to drug resistance and that exosomes from the resistant HMPOS-2.5R variant were able to transfer drug resistance to drug-sensitive HMPOS cells. When circulating exosomes from dogs with a favorable disease-free interval were compared with those from dogs with poor clinical outcome, a proteomic signature discriminating between the two cohorts could be identified, with several putative biomarkers shared with the aforementioned cell lines. This study highlighted the potential significance of exosomes in transferring drug resistance in canine osteosarcoma and indicated novel candidate biomarkers, which may be considered to monitor treatment response through liquid biopsy with the aim to better personalize chemotherapy.

Liquid biopsy has been explored also in human HGOS as a non-invasive method of studying circulating elements [42]. Therefore, the isolation of circulating EVs and analysis of their cargo may become a promising approach to discover and validate new biomarkers that could be used to improve treatment efficacies.

Further studies are however warranted to define the actual impact of EVs and exosomes for preclinical and clinical drug resistance in HGOS. If their involvement in HGOS drug resistance will be confirmed, new therapeutic avenues based on the strategy of mimicking EVs with micelleplexes or lipid-based nanoparticles to deliver drugs or RNA molecules to overcome or reverse chemoresistance may be proposed to design innovative clinical trials [43,44]. However, it must be underlined that the clinical use of exosomes still encounters several limitations [45]. Future research in this field should thus be devoted to improving their production and storage conditions to prevent loss of function and assure the long-term safety of exosome-based therapies.

### 2.5. Non-Coding RNAs

Non-coding RNAs (ncRNAs) include a large number of RNA subclasses, which are not transcribed into proteins and are mainly involved in the regulation of gene expression. According to their length, they can be classified as small non-coding RNA (sncRNA), long non-coding RNA (lncRNA) and circRNA. sncRNAs include the categories of small interfering RNA (siRNAs), micro RNAs (miRNAs), transfer RNAs and some ribosomal RNA. In recent years, an increasing number of ncRNAs have been suggested to play a role in HGOS tumorigenesis, invasion, metastatic progression, apoptosis and also in drug resistance [27].

The most widely studied ncRNAs are miRNAs and lncRNAs. miRNAs are single-stranded RNA fragments that repress target messenger RNAs (mRNAs) and play significant roles in multiple cellular processes [46]. Recently, several studies have indicated their involvement also in HGOS drug resistance [27,47,48]. Since this subject has recently been reviewed extensively [48], we have commented here on only a few reports, which highlight particular aspects of miRNA involvement in HGOS drug unresponsiveness.

Lin and coworkers [49] observed a correlation between miR-184 and doxorubicin resistance in HGOS cells. In particular, this study showed that in HGOS cells, elevated levels of miR-184 decreased doxorubicin sensitivity by targeting Bcl-2-like protein 1 (BCL2L1), an antiapoptotic protein, which promotes cell survival facilitating the acquisition of drug resistance.

Down-regulation of miR199a has been revealed in cisplatin-resistant HGOS cells and clinical samples [50]. This study demonstrated that overexpression of miR199a reversed cisplatin resistance targeting the hypoxia-inducible factor 1-alpha (HIF-1α), a transcription factor involved in the cellular adaptation to hypoxia [50].

Another study focused on the role of miR29 family for methotrexate resistance in HGOS [51]. The miR-29 family targets collagen type III Alpha 1 chain (COL3A1) and induced myeloid leukemia cell differentiation protein (MCL1). miR-29a, miR29b and miR29c were found to be downregulated in two methotrexate-resistant HGOS cell lines. In agreement with this observation, this study showed that overexpression of miR29a, miR29b and miR29c was able to sensitize methotrexate-resistant human HGOS cell lines to this drug by promoting apoptosis through the regulation of COL3A1 or MCL1.

In general, the so far reported body of evidence has indicated miRNAs as candidate predictive biomarkers of poor drug response, with the possibility to be taken into consideration also as therapeutic targets to reverse drug resistance, once their role in HGOS is confirmed and validated.

lncRNAs are RNA molecules more than 200 nucleotides in length, which play regulatory roles in different biological processes and diseases by interacting with DNA, RNA and proteins [52]. Several lncRNAs have been found to be aberrantly expressed in HGOS, showing some correlation with clinical outcome, disease status and drug resistance [27,48,53]. A list of the lncRNAs with the most relevant impacts on drug resistance, together with their mechanism of action, is presented in Table 1.

In general, the number of lncRNAs which may be involved in HGOS drug resistance is continuously increasing [27,48]. Indeed, Zhu and coworkers [70], by studying three sets of HGOS doxorubicin-resistant MG63/DXR cells in comparison with the parental MG63 cell line, identified 3465 lncRNAs (1761 up- and 1704 down-regulated) that were aberrantly expressed in resistant cells. On the same experimental models, these authors also found 3278 miRNAs that were either up- (1607) or down-regulated (1671), further supporting the deep involvement of different categories of ncRNAs in HGOS drug resistance which, once sufficiently validated, may become new biomarkers to predict patients’ drug responsivity.

An example of the crucial role of lncRNAs in drug resistance is the evidence that lncRNA FOXC2 antisense RNA 1 (FOXC2-AS1) can enhance the expression of FOXC2, subsequently increasing the ABCB1 levels and leading to doxorubicin resistance [67]. 

Another lncRNA involved in doxorubicin resistance in HGOS is the osteosarcoma doxorubicin resistance-related up-regulated lncRNA (ODRUL), which was found to be the most up-regulated lncRNA in doxorubicin-resistant HGOS cells [68]. Moreover, this study showed that lncRNA ODRUL was increased in HGOS patients with poor chemotherapy response and lung metastasis, and that its inhibition could decrease OS cell proliferation, migration, and partly reverse doxorubicin resistance in vitro. Since ABCB1 gene expression was also decreased after lncRNA ODRUL knockdown, the authors concluded that lncRNA ODRUL may act as a pro-doxorubicin-resistant molecule through the induction of ABCB1 overexpression in HGOS cells [68].

An interesting mechanism that has recently been described to play a key role in HGOS drug resistance is the so-called sponging activity of some lncRNAs (Figure 1), which bind miRNAs suppressing their regulatory functions [71]. For example, Fu and coworkers [56] found that the lncRNA TTN-AS1 is highly expressed in HGOS, in which it inhibits apoptosis and induces drug resistance by targeting miR-134-5p promoting expression of the malignant brain tumor domain 1 (MBTD1), a gene belonging to the polycomb family, involved in transcription processes. In agreement with this observation, MBTD1 was found to be correlated with a poor prognosis in HGOS patients [56].

Some lncRNAs have also been found to modulate the expression of drug efflux proteins belonging to the ABC transporter family. For example, it was reported that in HGOS cells, NORAD can induce overexpression of ABCC1 by suppressing miR-137 through a sponging mechanism [66].

In another study, the lncRNA ROR was found to be upregulated in cisplatin-resistant HGOS tissues and cell lines, in which it increased ABCB1 expression by sponging its negative regulator miR-153-3p [58].

### 2.6. Cancer Stem Cells

Tumor drug resistance may also be ascribed to subpopulations of cells with stemness properties and, consequently, termed cancer stem cells (CSCs). These cells can divide both into other CSCs, replenishing their own pool, and into cancer cells with different characteristics, increasing the intratumor heterogeneity [72]. CSCs were first identified in bone sarcomas by Gibbs and colleagues (2005) [73], who described their presence in both HGOS human specimens and cell lines. These cells were able to form spherical colonies called “sarcospheres” in non-adherent serum-free conditions and displayed markers associated with mesenchymal stem cells (MSCs) and embryonic cells [73]. However, the detection of CSCs in HGOS remains elusive, and putative candidates could be identified among undifferentiated MSCs or more committed osteoprogenitor cells, which underwent de-differentiation upon loss of p53 and retinoblastoma (Rb) genes [74].

More convincing evidence reported so far on HGOS CSCs indicated that their phenotype was characterized by increased expression of stemness-related factors, such as SRY (sex determining region Y)-box 2 (Sox2) [75], Kruppel-like factor 4 (KLF4) [76], octamer-binding transcription factor 3/4 (Oct3/4) and homeobox protein Nanog [77]. Among these, Sox2 was suggested as the most relevant marker of sarcoma CSCs, being crucial in sarcomagenesis [78]. 

One of the pivotal features displayed by HGOS CSCs is chemoresistance, which is frequently associated with increased metastatic ability [75]. This chemoresistant phenotype may derive from different mechanisms (Figure 2), among which the overexpression of ABC transporters [79], enhanced DNA repair activities [80], and altered modulation of apoptosis-related genes [81] can be mentioned. For instance, Roundhill and coworkers [82] recently indicated that HGOS CSCs exhibit elevated levels of ABCB1 and ATP binding cassette subfamily G member 1 (ABCG1), which may confer to these cells resistance against different drugs used for HGOS treatment such as doxorubicin, etoposide, vincristine, and actinomycin D.

Lee and colleagues [83] reported that MDR1 (encoding for ABCB1) and DHFR genes were upregulated in HGOS stem-like cells, which were respectively resistant to doxorubicin and methotrexate. 

Drug resistance of HGOS CSCs may also be influenced by epigenetic factors, which can modulate the expression profile of these cells. Di Fiore and colleagues [84] performed a molecular and genetic characterization of the 3AB-OS CSC line, which was derived from the MG-63 human HGOS cell line, and identified 189 miRNAs which were differentially expressed compared to parental cells, of which 37 were expressed only by the 3AB-OS cells. Among these, miR-29b-1 overexpression was found to be correlated with drug resistance [85]. Moreover, in another study performed on the same cell line, overexpression of miRNA let7-d was found to confer resistance towards several drugs [86], further confirming the role of miRNAs in acquisition of drug resistance by CSCs.

Finally, alterations of several signaling pathways have been implied in the origin and maintenance of HGOS CSCs subpopulations, despite not being directly associated with drug resistance. They include mitogen-activated protein kinase (MAPK) [87], Notch [88], Hedgehog [89], Wnt/β-catenin [90], bone morphogenetic protein 2 (BMP-2) [91], transforming growth factor beta 1 (TGF-β1) [92] and tumor necrosis factor alpha (TNFα) [93]. These factors and pathways are of interest because they may be considered as potential targets for therapy to prevent or inhibit the CSC phenotype. Indeed, the Wnt pathway was efficiently targeted by a tankyrase inhibitor in preclinical models of HGOS cancer stem-like cells [94], whereas acquisition of HGOS CSC phenotype was prevented by inhibition of the TGFβ pathway via single-walled carbon nanotubes [95]. 

CSCs may also be key players in microenvironment-related conditions that are important in determining the expression of several ABC transporters. Hypoxia, for instance, induces the HIF-1α transcriptional factor that up-regulates ABCB1 [96] and the activity of Notch homolog 1 (Notch1), which increases ABCC1 [97] thus inducing resistance to doxorubicin and methotrexate. Since active Notch1 was often found in aggressive and chemoresistant HGOS rich of CSCs [98], it cannot be excluded that the hypoxic niche selects CSC-like cells, which are by nature enriched of ABC transporters expression. 

Indeed, HGOS cells isolated as side-population of CSC-like elements display high expression of ABCB1, ATP binding cassette subfamily B member 2 (ABCB2), ABCG2 [78] and ATP binding cassette subfamily B member 5 (ABCB5) [99,100,101]. Notably, doxorubicin and, to a lesser extent, cisplatin and methotrexate, increases typical stemness markers including aldehyde dehydrogenase 1 family member A1 (ALDH1A1), Sox2, Oct4 and the Wnt/β-catenin pathway, which in turn up-regulates ABCB1 and ABCG2 [102]. Therein, chemotherapy may create a vicious circle that progressively selects CSCs with a more aggressive and chemoresistant phenotypes. This is consistent with other findings demonstrating that the selection in doxorubicin containing culture medium increases the proportion of HGOS cells with self-renewal capacity and high levels of ABCB1 [82], likely mimicking a process occurring during chemotherapeutic treatment in unresponsive patients.

## 3. Emerging Candidate Therapeutic Targets and Treatment Modalities

### 3.1. Chemorevertants and Emerging Modalities to Overcome Drug Resistance in HGOS

The most frequently studied approaches to circumvent ABC-mediated drug resistance have been based on the co-administration of chemotherapeutic drugs, which are substrates of these transporters, with compounds inhibiting ABC activity. During the last 30–40 years, different generations of ABC inhibitors have been developed, in order to obtain agents with higher efficacy and specificity together with fewer adverse toxicities and lack of antagonistic pharmacokinetic interactions with conventional chemotherapeutic drugs [97]. Unfortunately, when translated to clinical practice, these agents have shown limited therapeutic potential, mostly because of the severe toxic side effects observed at the concentrations required to significantly inhibit ABC transporter activity [103,104].

The most commonly studied ABC transporter is ABCB1, for which hundreds of biologically active compounds have been reported to act as its substrates and/or inhibitors. By taking into consideration the aforementioned clinical relevance of this transporter for HGOS, several strategies for reversing and preventing drug resistance by targeting ABCB1 have been studied in HGOS experimental models. 

Recently, we provided evidence that the newly generated ABCB1/ABCC1 inhibitor CBT-1^®^ (Tetrandrine, NSC-77037) may be considered as a potential adjuvant to standard chemotherapy in ABCB1-overexpressing HGOS patients (Figure 3) [105]. On the basis of these and other preclinical findings, an ongoing, recruiting phase I trial (ClinicalTrials.gov identifier: NCT03002805) is evaluating the clinical efficacy of the combination of CBT-1^®^ and doxorubicin for the treatment of metastatic, unresectable sarcoma (including HGOS) patients, who have progressed after treatment with doxorubicin.

Natural compounds or plant extracts have also been reported to act as potent ABCB1 inhibitors, exhibiting low collateral toxicity and good oral bioavailability [106]. Among these, curcumin was included in a Phase I/II clinical trial for relapsed or metastatic high-grade OS patients (ClinicalTrials.gov identifier: NCT00689195), for which however no results have been posted yet, despite its estimated completion date being set for June 2013. The rational of this approach was based on the fact that curcumin, a phenolic compound used in traditional Indian and Asian medicine, proved to inhibit several ABC transporters, including ABCB1 [107]. We also demonstrated that curcumin was able to decrease the ABCB1 transport activity in doxorubicin-resistant human HGOS cell lines, even if with a remarkably lower efficiency compared to other ABCB1 inhibitors [105]. Unfortunately, due to the lack of clinical information, it is not possible to indicate whether curcumin or curcuminoids might be of interest for HGOS treatment.

An alternative way to interfere with ABCB1 activity has emerged from studies on tyrosine kinase inhibitors (TKIs), which can be at the same time substrates but also down-regulators of ABC transporters activity by interfering with ATP-binding, therefore acting as chemoresistance revertants [108,109,110].

We have provided evidence that TKIs can be both ABCB1 substrates but also act as down-regulators of ABCB1-mediated resistance in HGOS experimental models. Indeed, we showed that the Aurora kinase inhibitor VX-680 (MK-0457, Tozasertib) (Figure 3) was less active in doxorubicin-resistant, ABCB1-overexpressing human HGOS cell lines but its combined administration together with doxorubicin proved to partially overcome resistance, most probably because the simultaneous presence of two ABCB1 substrates prevented this transporter from sufficiently extruding both drugs from the intracellular compartment [111]. Therefore, despite the fact that overexpression of ABCB1 and other ABC transporters can confer resistance to TKIs, thus limiting their use as single agents for cancer treatment [108,110], the association of these agents with conventional drugs, such as doxorubicin, may open new perspectives to overcome drug resistance in chemotherapy unresponsive patients.

Overcoming drug resistance may also take advantage of nanomedicine and studies on polymeric micelles, which may be considered for the targeted transportation of poorly water-soluble drugs or agents. The possibility of using polymeric micelles encapsulating miRNAs to actively target HGOS cells and overcome multidrug resistance has recently been reviewed [112]. Once candidate miRNAs and ncRNAs that have been indicated to be associated with drug resistance in HGOS will be sufficiently validated, this approach may be proposed as a novel therapeutic strategy.

If the role of EVs and exosomes in transferring drug resistance as described above is confirmed and validated, two possible additional treatment strategies to overcome resistance may be considered. The first one might be based on the inhibition of EV secretion by HGOS cells, but this approach is presently limited by the lack of agents that can specifically target EV secretion by cancer cells [34]. The second possibility might be the specific removal of tumor-derived EVs involved in the transfer of drug resistance, an approach that is presently under preclinical evaluation and has the advantage of not interfering with the normal secretion of “beneficial” EVs [34].

Additional emerging therapeutic strategies have been recently reviewed and are therefore not discussed in detail in this review [1,113].

### 3.2. Modified Conventional Drugs to Overcome Resistance in HGOS

Modification of already used chemotherapeutic drugs to produce synthetic agents exploiting specific metabolic vulnerabilities of resistant HGOS cells can be considered a promising chemosensitizing strategy. For instance, we found that U-2OS cell sublines with progressive increase in resistance to doxorubicin, also have a progressively increased mitochondrial mass and energetic metabolism based on tricarboxylic acid (TCA) cycle, fatty acid β-oxidation (FAO) and electron transport chain (ETC) [114]. A doxorubicin conjugated with a tripeptide vectorizing the drug toward mitochondria (Figure 3) dramatically reduced the number of mitochondria and induced damages in the mitochondrial DNA, impairing the transcription of several genes encoding energetic metabolism [115]. Such mitochondrial catastrophe dramatically reduces the oxidative-phosphorylation (OXPHOS)-linked production of ATP, increases the amount of reactive oxygen species (ROS) of mitochondrial origin and induces mitochondrial depolarization, triggering apoptosis. In our experimental models, these events were more pronounced in HGOS resistant cells, likely because they rely more on aerobic mitochondrial metabolism than sensitive cells, and were translated into the rescue of in vivo doxorubicin efficacy [114]. Since the heart is the main tissue damaged by doxorubicin and strongly relies on aerobic mitochondrial metabolism, the risk of such an extremely effective drug is to worsen the cardiac damage elicited by doxorubicin. Curiously, although the mitochondrial targeting doxorubicin reduced mitochondrial mass and energy metabolism in heart, normal cardiomyocytes showed a tremendous ability to compensate by promptly increasing the mitochondrial mass, differing from cancer cells [116]. Indeed, mice treated with this synthetic doxorubicin presented lower levels of creatine phosphokinase-MB (CPK-MB) than animals treated with parental doxorubicin [116]. Overall, these data suggest that mitochondrial targeting doxorubicin could be a promising compound to be tested in phase I clinical trials.

A second synthetic doxorubicin that may act with a similar mechanism is a nitrooxy-conjugated doxorubicin (Figure 3), a prodrug that, thanks to its high lipophilicity, is accumulated within mitochondria, where it is broken into doxorubicin and nitric oxide (NO) [117]. In mitochondria, NO inhibits aconitase, a critical enzyme of TCA cycle, and the Fe-S-cluster containing proteins of the ETC, triggering a strong mitochondrial-dependent apoptosis [117]. Although not yet tested in HGOS, the liposomal formulation of this nitrooxy-doxorubicin demonstrated a good efficacy in doxorubicin-resistant triple negative breast cancer, coupled with a lower cardiotoxicity compared to native doxorubicin [118]. Hence, nitrooxy-doxorubicin is a new agent to be studied in HGOS in order to verify whether it might represent a new option for the treatment of HGOS refractory to doxorubicin. This approach might also have an additional advantage since it has been demonstrated that NO can deplete GSH, thus reducing cisplatin inactivation and enhancing its activity [119].

Another key difference between doxorubicin-sensitive and -resistant cells is the differential response to endoplasmic reticulum (ER) stress, which is impaired in HGOS and other cancer cells with acquired resistance to doxorubicin [120,121]. When proteins are folded within ER lumen, they are physiologically subjected to a quality control, carried out by the ER-associated protein degradation/ER-quality control (ERAD/ERQC) machinery associated to ER membrane. If the protein is correctly folded, it is delivered to its final destination; on the contrary, if it has misfolded/unfolded portions it is extracted, ubiquitinated and primed for proteasomal degradation [122]. Interestingly, HGOS cells with increased resistance to doxorubicin up-regulate several proteins of the ERAD/ERQC complex [123], likely because the constant pressure exerted by the drug in the culture medium stimulates cells to deal with damaged proteins, which must be removed to prevent the activation of ER stress-dependent apoptotic pathways. ERAD/ERQC complex, however, works at its maximal efficacy in resistant HGOS cells and can be easily overwhelmed by perturbing agents. To this aim, we designed a synthetic doxorubicin releasing H2S (Figure 3), devoid of any oxidative damaging effects on cardiomyocytes but still able to kill doxorubicin resistant osteosarcoma cells [124]. The drug, indeed, is accumulated within the ER where it sulfhydrates several nascent proteins, accumulating the burden of unfolded proteins and activating the CCAAT/enhancer binding protein-β liver inhibitory protein (C/EBP-β LIP)/C/EBP-β homologous protein (CHOP)/p53 upregulated modulator of apoptosis (PUMA)/caspases 12-7-3 pro-apoptotic axis [123]. Similarly to mitochondrial targeting doxorubicin, the maximal efficiency of H2S-releasing doxorubicin is achieved in resistant cells, because their ERAD/ERQC system is already saturated under basal conditions and collapses upon any stimulus, further increasing the burden of misfolded proteins. Notably, ABCB1 is also folded within the ER and is sulfhydrated by H2S-releasing doxorubicin: this process alters the stable conformation of ABCB1 [125], determining instead a prominent ubiquitination of the protein [123]. This provides an additional mechanism explaining the efficacy of H2S-releasing doxorubicin in murine doxorubicin resistant HGOS models [126], with perspective of a possible clinical translation.

Platinum-modified compounds have also been studied and some preclinical findings have been reported also for HGOS cells. For example, diaminedichloro-platinum (II) complex and camptothecin dual compounds proved to overcome cisplatin resistance in the U2OS/Pt HGOS cell line [127]. Other promising compounds, which have recently shown interesting preclinical efficacy in HGOS experimental models, include bifunctional platinum(II) complexes with bisphosphonates (showing high affinity for hydroxyapatite) [128,129] and platinum complexes containing 8-hydroxyquinoline ligands [130].

Recently, photodynamic therapy has become a new approach to achieve a spatial-temporal control of tumor killing. In the case of HGOS, the first in vitro assays based on the combination of cisplatin and low-level laser, which enormously amplifies the ROS generated by chemotherapy, have been successful in reducing tumor cell viability [131]. Thanks to the high degree of control in terms of killing area, amount and temporal release of ROS, this photochemotherapeutic approach can be a new promising option to precisely reduce the tumor mass during surgery or eradicate accessible metastasis refractory to chemotherapy.

An alternative approach considered the use of MSCs loaded with photosensitizer-coated fluorescent nanoparticles and the photosensitizer meso-tetrakis (4-sulfonataphenyl) porphyrin, which were co-cultured with a human HGOS cell line [132]. After irradiation with light, the release of reactive oxygen species (ROS) was able to trigger cell death of HGOS cells, proving the efficacy of this approach.

The preclinical research on modified conventional new drugs or physical agents may set the basis to improve the clinical treatment of HGOS in the near future.

### 3.3. Nanocarriers and Nanoparticles

Delivery of chemotherapeutic agents by functionalized nanocarriers has been indicated as an effective strategy to protect drugs from rapid clearance, prolong their circulating time, and increasing their concentration at tumor sites, thus enhancing therapeutic efficacy and reducing side effects [133]. 

In recent years a lot of work has been done to formulate and test nanoparticles carrying conventional drugs used in the chemotherapeutic treatment of HGOS.

Proteins or molecules that are overexpressed on HGOS cells’ surface have been considered as ligands to interact with nanoparticles and to promote their internalization into cancer cells, with a consequent anticancer effect [134].

Following the first clinical trial using Caelyx/Doxil, which lead to only modest outcome improvements of HGOS patients [135], new liposomal formulations have been prepared to increase the efficacy of liposomal doxorubicin.

The thermo- and pH-sensitive controlled release of doxorubicin from liposomal formulations [136] or the active targeting of surface antigens, as anti-activated leukocyte adhesion molecule (ALCAM)/CD166) [137] or anti-ephrin alpha 2 (EPHA2) receptor [138], have improved the efficacy of liposomal doxorubicin compared to untargeted liposomes or free drug.

Recently, the hyaluronic acid (HA) receptor CD44 has been found to be overexpressed in HGOS cells and indicated as a new attractive candidate target. Our group has recently validated the safety and efficacy of HA-liposomal doxorubicin and H2S-releasing doxorubicin against murine drug resistant HGOS [126]: both formulations have shown higher efficacy than Caelyx. Part of this effect could be due to the decreased expression of ABCB1 induced by HA [139] or by the ubiquitination of this transporter in case of H2S-releasing doxorubicin [126]. Moreover, the CD44-triggered endocytosis increases the intracellular delivery of doxorubicin in HA-conjugated liposomes [126].

To further improve the efficacy of doxorubicin against drug resistant HGOS, codelivery strategies of doxorubicin plus a sensitizing agent were further developed. For instance, the EPHA2-receptor targeting liposomal doxorubicin was co-loaded with a siRNA specific for the JNK-interacting protein 1 (JIP1) [140], a protein involved in doxorubicin resistance [141]. In these dual target liposomes, the siRNA did not interfere with the release of doxorubicin but, at the same time, targeting the EPHA2-receptor increased drug delivery and cytotoxicity in HGOS cells [140].

Recently, melatonin has been shown to exert a pro-apoptotic effect against HGOS, likely by down-regulating the X-linked inhibitor of apoptosis (XIAP), survivin and human telomerase catalytic subunit (hTERT) [142]. The codelivery of doxorubicin and melatonin, achieved through graphene dendrimers-Fe3O4 nanocarriers, proved to enhance the apoptosis elicited by doxorubicin in HGOS cells and human bone-marrow-mesenchymal stem cells [142].

The use of molecules with high affinity for bone is another strategy that has been explored to deliver drugs into HGOS. Following this rationale, micelles made of hydrophilic D-aspartic acid octapeptide and 11-aminoundecanoic acid have shown a prompt absorption on hydroxyapatite and a good pH-dependent delivery of doxorubicin in Saos-2 cells [143]. Similarly, a hydroxyapatite-doxorubicin conjugate has recently been produced and coated with the poly(lactide-co-glycolide) (PLGA) polymer: this conjugate is effective in vitro [144], but its pharmacokinetic and pharmacodynamic profile has not been explored in vivo.

CaCO3-methoxy poly(ethylene glycol)-block-poly(L-glutamic acid) nanoparticles loaded with doxorubicin exploit the high affinity of the nanocarrier for mineralized tissues, and display higher delivery of doxorubicin, higher anti-tumor efficacy and lower side-effects than doxorubicin in mice HGOS [145].

Finally, taking advantage of the high bone tropism of aminobisphosphonate, doxorubicin loaded within poly-electrolyte, poly(acrylic acid) functionalized alendronate [146], or carried by alendronate/low molecular weight heparin-decorated liposomes [147], have been successfully tested against HGOS xenografts, revealing higher efficacy than the free drug.

With a combination of multiple technological strategies, liposomal doxorubicin decorated with HA and alendronate has been co-administered with the tumor penetrating peptide RGD in mice bearing orthotopic HGOS: the sum of high bone tropism induced by alendronate, increased EPR mediated by RGD, and CD44-triggered endocytosis of doxorubicin enormously enhanced the effects of the drug in these preclinical models [148].

The liposomal encapsulation of other drugs used as front-line treatment, such as cisplatin and methotrexate, is more difficult, because the physico-chemical structure of these chemotherapeutics impairs significant encapsulation of the compounds in the liposomal bilayer or in the aqueous core. Indeed, the validation of liposomal nedaplatin is very recent and has shown a discrete efficacy in vitro against U-2OS cells [149].

Nanocarriers other than liposomes, adopting the same strategies used for the active targeting of doxorubicin, have been successfully developed for these two drugs, such as cisplatin-loaded calcium phosphate nanocomposites [150], cisplatin-loaded graphene oxide-hydroxyapatite nanoparticles coated with chitosan [151], cisplatin-bisphosphonate conjugates [129], and meso-porous zinc-substituted hydroxyapatite decorated with methotrexate [152]. The cisplatin-bisphosphonate conjugates appeared to be the most promising agents, since they showed increased anti-tumor efficacy and reduced systemic toxicity in animal models [129].

One major limitation of most studies performed on nanocarriers is that they have shown to increase the effects of chemotherapeutic drugs against commercially available cell lines with low expression of ABC transporters. Only a few studies have tested nanocarriers against drug-resistant cells, e.g., in cells overexpressing ABCB1. The investigations on drug-resistant HGOS xenografts will be a critical issue in validating the large series of formulated nanocarriers to define whether they can be proposed as valid tools to reverse resistance to conventional chemotherapy.

Since the standard treatment regimen is based on polychemotherapy, a second critical issue is to determine if nanoparticle-loaded drugs impact on the efficacy of the co-administered chemotherapeutics. Only one study demonstrates that the combination of cisplatin-loaded nanoparticles and doxorubicin has synergistic effects in vitro [153]. The presence of unexpected drug–drug interactions or undesired in vivo side effects has not been investigated yet, and is a crucial point in determining the feasibility of nanocarrier-based strategies in patients.

A second study, using hydrogel formulations co-loaded with doxorubicin, cisplatin and methotrexate, showed that the hydrogel effectively reduces cell proliferation and tumor growth of MG-63 and Saos-2 HGOS xenografts, achieving the complete release of the drugs from hydrogel 11 days after the administration and a good control of tumor growth until day 16, without signs of organ toxicity [154]. The main limitation of this fascinating approach is the need for using an intratumor injection of the hydrogel to reach a high concentration of the chemotherapeutic drug. The efficacy and toxicity after systemic administration of the hydrogel-based system have not been documented.

Overall, although nanotechnology may offer a variety of advantages compared to conventional chemotherapy, the feasibility, safety and efficacy of most nanocarrier-based approaches must be robustly validated in different preclinical models of HGOS before proceeding to the first clinical trials.

## 4. Conclusions

Drug resistance is a multifactorial phenomenon in HGOS, as well as in several other human cancers, and validation of newly indicated chemoresistance-related factors, which may also act as novel therapeutic targets, is a major challenge to improve effectiveness of chemotherapy together with a personalization of treatment regimens.

Different approaches, such as pharmacogenomics, genomics, epigenetics, transcriptomics, or proteomics may be combined to define genetic differences which define individual drug metabolism and responsiveness, leading to the final treatment response of each individual patient.

The recent development of advanced genomic and proteomic technologies has significantly improved drug targets and biomarkers discovery, providing information underlying the heterogeneous response to anticancer drugs of each individual patient, which is mandatory for planning personalized treatments. However, it must be considered that these new powerful sequencing technologies also have some challenges, mainly due to the existence of intron–exon, repetitive sequences, and pseudogenes that may reduce clinical annotation. Moreover, non-coding RNAs, which are increasingly emerging as drug resistance determinants, may not be efficiently detected by all technologies.

Proteomics studies will also be crucial in validating any drug resistance-related gene expression alteration, in order to indicate biomarkers which may be considered as new therapeutic targets.

The hoped transfer of novel drug resistance-related biomarkers identified through this integrated approach into clinical practice should greatly benefit future drug development for tailored therapies aimed to overcome HGOS drug resistance.

## Figures and Tables

**Figure 1 cancers-13-02878-f001:**
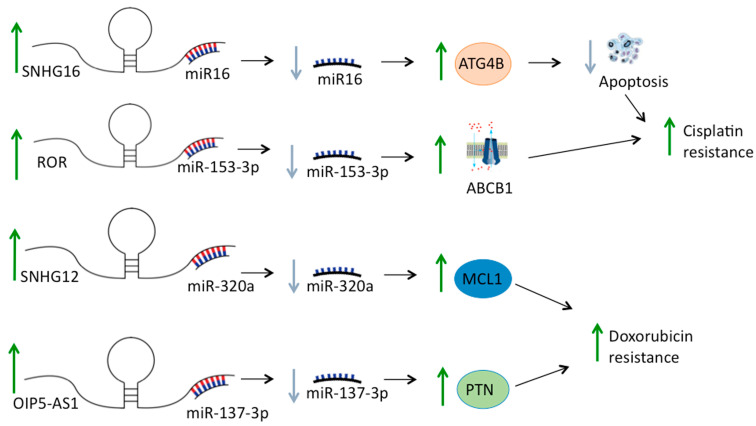
Examples of sponging activity of lncRNAs with impact in osteosarcoma drug resistance. Through sponging activity, lncRNAs can regulate mRNA expression by competitively binding complementary miRNAs and, consequently, opposing their interaction with target mRNAs. Figure shows some lncRNAs that are overexpressed in osteosarcoma cells and negatively modulate the expression of target genes by sponging their regulatory miRNAs.

**Figure 2 cancers-13-02878-f002:**
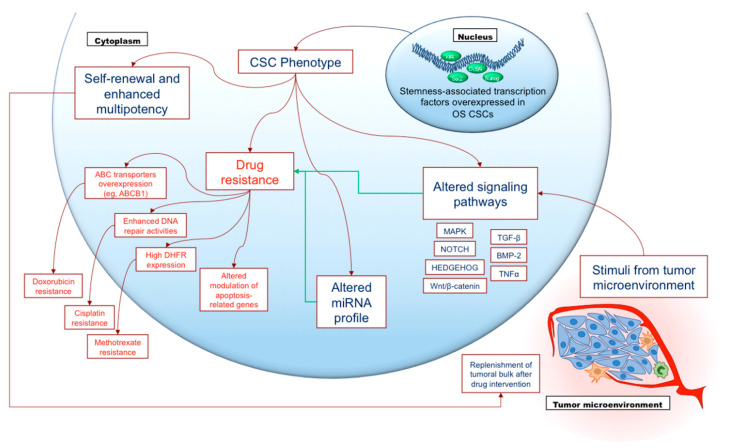
Involvement of osteosarcoma cancer stem cells in drug resistance. Osteosarcoma cancer stem cells (OS CSCs) represent a relatively small subpopulation in the tumoral bulk, characterized by high expression of stemness-related transcription factors (such as Sox2, Oct3/4, Nanog and Klf4). These characteristics promote self-renewal capability and enhanced multipotency, which enable OS CSCs to persist and to generate neoplastic cells as well, replenishing the tumoral bulk, i.e., following chemotherapy or surgical interventions. Nonetheless, the most critical feature of OS CSCs is enhanced drug resistance towards the main anti-neoplastic agents involved in high-grade osteosarcoma treatment, as a consequence of different mechanisms (the most relevant of which are shown in this Figure). Such chemoresistance may be enforced by aberrant miRNA expression and by alteration of different signaling pathways mainly induced by stimuli of the tumoral microenvironment.

**Figure 3 cancers-13-02878-f003:**
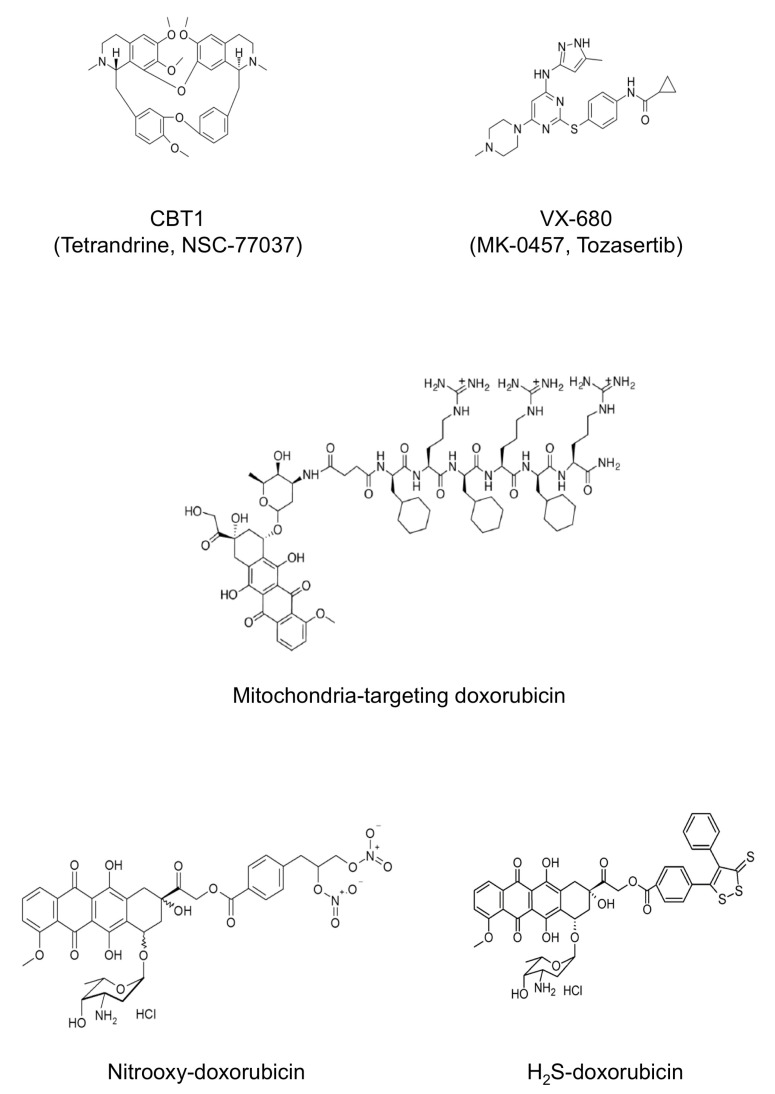
New candidate agents to overcome drug resistance in HGOS. The chemical structures of new drugs discussed in chapter 3. Only molecules which are not protected by copyright are shown.

**Table 1 cancers-13-02878-t001:** List of non-coding RNAs indicated to be involved in osteosarcoma drug resistance.

Name	Expression	Mechanism	References
**LncRNA Targeting miRNAs**			
LINC00161	Down-regulated	Promotes apoptosis by sponging miR-645 and upregulating IFIT2	[54]
SNHG16	Up-regulated	Increases cisplatin-resistance upregulating ATG4B by sponging miR-16	[55]
TTN-AS1	Up-regulated	Increases cisplatin-resistance promoting MBTD1 expression by targeting miR-134-5p	[56]
NCK-AS1	Up-regulated	Increases cisplatin-resistance upregulating MRP1 by targeting miR-137	[57]
ROR	Up-regulated	Increases cisplatin-resistance upregulating ABCB1 by targeting miR-153-3p	[58]
SNHG12	Up-regulated	Increases doxorubicin resistance promoting the expression of MCL1 by targeting miR-320a	[59]
LUCAT1	Up-regulated	Increases methotrexate resistance upregulating ABCB1 by targeting miR-200c	[60]
NEAT1	Up-regulated	Increases cisplatin resistance sponging miR-34c	[61]
SARCC	Down-regulated	Increases cisplatin sensitivity promoting miR-43 expression, promoting down regulation of Hexokinase 2	[62]
CTA	Down-regulated	Increases Doxorubicin sensitivity promoting apoptosis by binding miR-210 and inhibiting autophagy	[63]
OIP5-AS1	Up-regulated	Increases doxorubicin resistance upregulating PTN by targeting miR-137-3p	[64]
MIR17HG	Up-regulated	Increases cisplatin resistance suppressing miR-130-3p and upregulating SP1	[65]
NORAD	Up-regulated	Increases cisplatin resistance targeting miR-410-3p	[66]
**LncRNA Targeting ABC Transporters**			
FOXC2-AS1	Up-regulated	Increases doxorubicin resistance upregulating ABCB1 by increasing FOXC2	[67]
ODRUL	Up-regulated	Increases doxorubicin resistance increasing ABCB1 expression	[68]
FENDRR	Down-regulated	Increases doxorubicin sensitivity promoting apoptosis and down regulating ABCB1 and ABCC1	[69]

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
