# Peer review of "Drug Resistance in Osteosarcoma: Emerging Biomarkers, Therapeutic Targets and Treatment Strategies"

_cancers, 2021, doi:10.3390/cancers13122878_

Round 1
Reviewer 1 Report
In the present formate paper is suitable for publication without further modification.
Reviewer 2 Report
I am satisfied with revised version of the article.
Reviewer 3 Report
This article written by Hattinger and colleagues is a resubmission of a review of literature previously rejected. The manuscript discusses a topic of great interest in the osteosarcoma (and wider cancer) field. My major concern was about a redundancy with another manuscript published in the same journal in February 2021 by Marchandet et al named “Mechanisms of Resistance to Conventional Therapies for Osteosarcoma”. The authors’ arguments are convincing, and they have changed their manuscript properly responding to this issue. They shortened the part about ABC transporter and extended the chapter "extracellular vesicles". They also modified figure 1 which is now more informative. Taking all these together, the authors have satisfactorily responded to all my questions and suggestions, and I can recommend this review of literature to go further.
This manuscript is a resubmission of an earlier submission. The following is a list of the peer review reports and author responses from that submission.
Round 1
Reviewer 1 Report
The manuscript by Hattinger and colleagues discuss a topic of great interest in the osteosarcoma (and wider cancer) field. Drug resistance is one of the most important processes causing failure of chemotherapies and decrease of patient survival. The authors described properly the different mechanisms responsible of chemoresistance in high-grade osteosarcoma and new treatment strategies.
I noticed some minor points in the manuscript that need to be clarified or improved:
- In chapter 2.2 “DNA repair factors”:
The authors explained that “APEX1 and PARP1 have been associated with reduced drug responsiveness and clinical outcome in HGOS”. I suppose it is correlated with a high gene expression but it needs to be clarified.
About the association of a PARP inhibitor with an ATR kinase inhibitor, it should be interesting to briefly described role of ATR kinase in DNA damages and cell cycle and explain the relevance of this combination in osteosarcoma.
The last two paragraphs about the GSH/GST systems are not strongly related to the chapter title. It is not a system directly involved in DNA repair and the authors should move this part in another chapter or rename the chapter.
- In chapter 2.4 “Extracellular vesicles”:
This chapter seems too superficial. The authors explained a role of extracellular vesicles in drug resistance without described any biological mechanisms. It will be interesting to add some clear examples from research articles.
- Figure 1
This figure is also superficial. It is a basic representation of a molecular mechanism with no information in chemoresistance of osteosarcoma. I would recommend adding some details about the role of sponging activity of lncRNAs in drug resistance.
Unfortunately, a large part of this review seems very redundant with another manuscript published in the same journal in February 2021 by Marchandet et al named “Mechanisms of Resistance to Conventional Therapies for Osteosarcoma”. My major concern is the real interest for scientific community of another similar review of literature. Nevertheless, the chapter 3 “Emerging candidate therapeutic and treatment modalties” is original and very accurate.
The manuscript is clear and well written but due to a recent publication about a similar topic, I cannot recommend this review to go further.
Reviewer 2 Report
The review article entitled "Drug Resistance in Osteosarcoma: Emerging Biomarkers, Therapeutic Targets and Treatment Strategies" by Hanttinger et al. is suitable for publication in Cancers journal after minor revision.
Please add some representative figures in the Consolidated and Emerging Drug Resistance-Related Biomarkers section.
Reviewer 3 Report
The authors have presented a very interesting and full review of the topic of Drug Resistance on Osteosarcoma. Some observations and conclusions based on the presented material are deep and should be taken into consideration. At the same time, I would like to ask them to present structures of the discussed drugs (especially small molecules) in the text. It would improve the visual presentation of presented material.